# Development of a personalized digital biomarker of vaccine-associated reactogenicity using wearable sensors and digital twin technology

Steven R. Steinhubl [1] ✉, Jadranka Sekaric[2], Maged Gendy[2], Huaijian Guo[3], Matthew P. Ward[1], Craig J. Goergen[1], Jennifer L. Anderson[1], Sarwat Amin[1], Damen Wilson[1], Eustache Paramithiotis[3] & Stephan Wegerich[2]

## Abstract

**Background** Effective response to vaccination requires activation of the innate immune system, triggering the synthesis of inflammatory cytokines. The degree of subjective symptoms related to this, referred to as reactogenicity, may predict their eventual immune response. However, the subjective nature of these symptoms is influenced by the nocebo effect, making it difficult to accurately quantify a person's physiologic response. The use of wearable sensors allows for the identification of objective evidence of physiologic changes a person experiences following vaccination, but as these changes are subtle, they can only be detected when an individual's pre-vaccination normal variability is considered. **Methods** We use a wearable torso sensor patch and a machine learning method of similarity-based modeling (SBM) to create a physiologic digital twin for 88 people receiving 104 COVID vaccine doses. By using each individual's pre-vaccine digital twin, we are able to effectively control for expected physiologic variations unique to that individual, leaving only vaccine-induced differences. We use these individualized differences between the pre- and post-vaccine period to develop a multivariate digital biomarker for objectively measuring the degree and duration of physiologic changes each individual experiences following vaccination. **Results** Here we show that the multivariate digital biomarker better predicted systemic reactogenicity than any one physiologic data type and correlated with vaccine-induced changes in humoral and cellular immunity in a 20-person subset. **Conclusions** A digital biomarker is capable of objectively identifying an individual's unique response to vaccination and could play a future role in personalizing vaccine regimens.

## Plain Language Summary

Researchers have created a new way to track how people's bodies react to COVID-19 vaccines using a small sensor patch. Normally, doctors rely on people describing their symptoms, but this can be affected by what they expect to feel. This new method uses artificial intelligence to first learn each person's normal body patterns before they get the vaccine. After vaccination, the sensor measures any small changes in the body and compares them to the person's usual patterns. In a study of 88 people, scientists found that combining different body measurements into one signal was better at predicting vaccine side effects and immune responses than looking at single measurements alone. This technology could help doctors better understand how each person responds to vaccines and make vaccination plans more personalized in the future.

Immune cell activation is essential to a successful vaccination strategy, with inflammation being the immune system's initial response to this activation. The level of vaccine-induced inflammation plays an essential role in its tolerability and potentially its efficacy[1,2]. Presently, the symptomatic tracking of reactogenicity—the physical manifestations of vaccine-related inflammation—is the only measure of acute vaccine response that is monitored at any scale[3]. For example, following the second dose of a mRNA based COVID-19 vaccine nearly 70% of individuals participating in the

Centers for Disease Control and Prevention's (CDC) v-safe study (<5% of all vaccinated individuals in the US) reported having a systemic symptom such as fatigue, myalgias or chills[4]. However, the subjective nature of these data limit their value as a measure of vaccine-induced inflammation as they are susceptible to a nocebo effect. An analysis of the placebo arms of COVID-19 vaccine randomized trials found that 76% of systemic symptoms experienced after the first dose, and 52% after the second dose could be attributed to the nocebo effect[5].

[1]Purdue University, W, Lafayette, IN, USA. [2]Prolaio Health, Scottsdale, AZ, USA. [3]CellCarta, Montreal, QC, Canada. ✉e-mail: ssteinhu@purdue.edu

Currently, there are no objective measures of reactogenicity routinely used. Measuring vaccine-induced inflammation has depended on intermittent and infrequent sampling of blood-based soluble factors such as chemokines and cytokines[6]. Most, but not all, of these studies have found significant relationships between blood-based inflammatory biomarkers and measures of reactogenicity and/or immunogenicity following vaccination against COVID-19 or other pathogens[1,2,7–9]. However, due to the invasive requirements of these studies, sample sizes are limited, and the frequency and duration of testing are minimized. Non-invasive methods of measuring changes in soluble inflammatory biomarkers following vaccination, such as through urine and saliva sampling, have been evaluated, but have yet to be proven effective [6,10].

A potentially novel approach to quantifying the totality of an individual's physiologic response to vaccination could be through wearable sensors that can continuously track individual physiologic and behavioral changes following vaccination to create a digital biomarker[11]. Recently, a range of wearable sensors—wrist wearables, rings and torso patches—have been shown to be able to detect the subtle physiologic changes following COVID-19 vaccination[12–18]. The degree of changes are so small that without knowledge of a person's unique pre-vaccine normal levels and natural variability, the detection of these subtle deviations would not be possible. Changes that, for example, might be only two beats per minute difference in resting heart rate—a change that would never be detectable on a population level due to marked inter-individual variability in resting heart rate[12,19]. As much of this prior work has utilized consumer devices, physiologic and behavioral changes following vaccination have mostly been determined based on a single daily summary value for each parameter, with most physiologic measures determined during sleep. Even with this limited data density, multiple studies have found significant associations between post-vaccine deviations in physiologic measures, subjective symptoms and humoral immune response [13,20,21].

In the present study, we sought to develop a personalized digital biomarker for COVID-19 vaccine-induced reactogenicity utilizing several advanced technologies; a medical grade patch biosensor able to continuously capture multiple biometrics, and an analytics platform using a machine learning method of similarity-based modeling (SBM), which learns the dynamic interplay between multivariate input sources[22]. Combined, these technologies enabled the development of personalized pre-vaccine baseline models of each participant's unique physiologic dynamics—a physiologic "digital twin."[23] When these models were then applied to post-vaccination data to remove expected individual variations, continuous vaccine-induced changes were able to be isolated to create a multivariate, personalized biomarker of reactogenicity—a multivariate change index of reactogenicity (MCIR).

Here we report the measured physiologic changes using this novel, individualized digital biomarker following vaccination against COVID-19 in 88 volunteers who received a total of 104 vaccines in a real-world setting. We quantify the relationship between that measure and subjective symptoms in all individuals, finding a statistically significant correlation that performed better than any single physiologic data type. In addition, in a subset of participants, we identified a statistically significant relationship between MCIR and the humoral and cellular immune responses early following vaccination. These results provide preliminary support for the potential value of a multivariate digital biomarker to objectively quantify the totality of individual reactogenicity following vaccination.

## Methods
### Clinical setting
Participants were recruited through one of two study protocols. The majority of participants were enrolled through the Vaccine-Induced Inflammation Investigation (VIII) study protocol. The second protocol was the Continuous Physiologic Monitoring for Immune Response via Wearable Sensor Data (COmMON SENS) study. The VIII study was approved by the Sterling IRB (ID# 8842-SRSteinhubl) in April of 2021. Clinicaltrials.gov registration number was NCT05237024. The COmMON SENS study was approved by the Purdue University IRB (ID# IRB-2021-453) April 2021.

For the VIII study, individuals who were already voluntarily planning to receive a vaccine against COVID-19 were recruited from the general population primarily via email outreach disseminated by employees of the study sponsor, physIQ (Chicago, IL, USA, but since purchased by Prolaio, Scottsdale, AZ, USA). Employees were encouraged to further disseminate recruitment information to their family and friends. The choice by any potential participant to get a vaccine was entirely voluntary and only people already planning to be vaccinated were approached to enroll in the study. Participants in the immunogenicity substudy of the VIII study were recruited in a similar manner with outreach disseminated by CellCarta (Montreal, Qc, CA), the cellular immunogenicity lab. All participants provided written informed consent prior to enrollment.

In the COmMON SENS study, Purdue University students, staff, and faculty were recruited via advertising to the general university population, affiliates, and friends. Again, only individuals who were voluntarily planning to receive a COVID-19 vaccine were recruited.

### Inclusion and exclusion criteria
Any individual over age 18 (participants between age 12 and 17 were allowed in the VIII study, although none were recruited) who was planning to receive any of the 3 FDA-approved vaccines against COVID-19 were eligible for enrollment. The only exclusion criterion was known allergy to the adhesive of the sensor patch.

### Study methods
Individuals enrolled were asked to wear a patch sensor for ~12 to14 days surrounding vaccination. Participants could agree to monitor themselves during more than one vaccine dose. Volunteers were asked to place the patch on themselves and begin monitoring 5 days prior to their planned vaccination and continuing for a total of up to 14 days. As the battery life of each disposable patch was ~7 days, each participant was asked to sequentially wear two patches at the time of each vaccine.

All participants received a locked-down Android phone with a pre-loaded app to enable patch and survey data capture. The app enabled participants to mark the day and time they received each vaccine dose and respond to daily survey questions for up to 7 days following vaccination to track all subjective symptoms and to document if they took any anti-inflammatory analgesic agent.

### Immunogenicity sub-study
Participants in the immunogenicity sub-study followed the identical protocol as all study participants with the exception of agreeing to a series of blood draws. For participants who had yet to receive their first vaccination ($n = 3$), there were 4 scheduled blood draws. The first ~5 days prior to their first vaccine, the second ~14 days after their first dose, the third ~14 days after their second dose, and their final blood draw ~60 days after their second vaccine dose. The majority of participants underwent only 3 blood tests - ~5 days before their second or third vaccine dose, and again ~14 days and ~60 days after that dose. Immune response was determined based on the change from the 5-day pre-vaccine level to the 14-day post-vaccine level.

### Wearable sensor
The VitalPatch™ by VitalConnect (San Jose, CA) is an FDA 510(k)-cleared, wearable, disposable adhesive patch with an integral one-time use battery and integrated electronics. The battery life of each patch lasts 7 days. The patch was self-applied by the participant to their left upper chest. Guidance for placement and confirmation of connectivity was provided via the app.

The patch transfers biosignal data over Bluetooth low-energy protocol to the mobile app. Once the mobile app has received the biosignals from the VitalPatch™, the biosignals are uploaded to a cloud-based server using digital cellular or Wi-Fi networks. No personal identifiers were stored or transmitted with the data either from the sensor or from the mobile app. Upload via digital cellular network was secured with Transport Layer

Security (TLS) cryptographic protocol between the mobile phones and the server. The platform was securely hosted in the Google cloud and the analytics server stored the raw physiological telemetry data captured by the study device. All the telemetry waveform data were stored only by participant ID. The data could only be obtained or viewed via secure authenticated login.

## Study platform and personalized physiology analytics

The cloud-based analytics platform used a general machine learning method of similarity-based modeling (SBM), to analyze collected data. SBM models the behavior of complex systems (e.g., aircraft engines, computer networks, or human physiology) by learning tandem patterns among system variables as they are periodically sampled together[22]. Personalized baseline models of each participant's unique physiologic dynamics are established, creating a "digital twin," which when compared to new input data following a possible immune-stimulator, removes expected variations, ideally then leaving only vaccine-induced differences. These differences are the residuals. These residuals are combined into an Multivariate Change Index of Reactogenicity (MCIR), which is updated on a 15-minute basis, allowing for the tracking of the onset, offset and degree of physiologic changes. Patch data was filtered for quality using an ECG-based signal quality index (SQI), that is graded 0-1 with 1 as highest quality.

**Uni-parametric analysis.** Patch ECG-derived cardiorespiratory features were filtered with a threshold of SQI > 0.9. Since activity level and skin temperature derived from the patch are not related to the ECG signal, they were not filtered with SQI, however any skin temperature values below 33 °C and above 42 °C were ignored. Five, 3-hour aggregate parameters were derived from 1-min derived features after baseline normalizing using z-scoring on an individual participant basis. The mean and standard deviations used for the z-scoring, were calculated from feature data generated during the 48-hour window prior to a vaccine dose. (Supplementary Fig. 1) The corresponding post vaccine dose aggregate data for each participant were used to identify cases where at least one standard deviation of change from baseline occurred.

**Multiparametric (MCIR) analyses.** The multivariate MCIR is a personalized modeling algorithm that uses coincident heart rate, heart rate variability, respiration rate, activity level and skin temperature derived from the patch device as input. A unique model was trained for every participant based on data collected prior to vaccine doses. For data to be used as input to MCIR, activity level had to be less than 0.05 g (a level corresponding to normal walking), skin temperature between 30 °C and 40 °C, and SQI ≥ 0.9. Additionally, during training, heart rate was required to be between 40 bpm and 250 bpm, and respiration rate between 8 and 35 breaths per minute. To train an individual MCIR model, 2500 one-minute samples of input variables were required to be available prior to vaccination and that the 1-min samples had to be distributed over 3 days.

**MCIR total response.** The metric defined for assessing MCIR total reactogenicity response ("MCIR Total Response"), employed an area under the curve (AUC) approach. The metric is defined for a fixed, 72 h, window of time starting from the time of administration of a vaccine dose. The MCIR Total Response was defined as $A_i/A_T$ as illustrated in Supplementary Fig. 2, where $A_T$ is the total rectangular area within the time window and $A_i$ is the area under the curve for MCIR during the window.

**MCIR detectable response.** A detectable response was defined as a collection of MCIRs with persistent, non-zero MCIR values within the fixed 72-h window following a vaccine. The 72-h window was selected to capture as much of the vaccine-induced physiologic response and minimize any potential noise. In addition to being a persistent trend

(non-zero MCIR trend > 1 h), the MCIR response had to satisfy conditions that were designed to rule-out small magnitude, random fluctuations that are likely due to noisy arbitrary inputs. Detectable response was defined as the presence of at least 50% of 15-minute steps within a 6-hour sliding window with an MCIRs > 0.10. If any of such occurrences has been found in the fixed 72-hour window following a vaccine the individual has a detectable physiologic change. In this way we identified 66 (63.5%) vaccine doses for which an identifiable physiologic deviation from their expected 'normal' was of detectable size. To estimate the false positive rate of defining "detectable" as above, we selected an independent dataset consisting of 76 participants from two different, healthy, non-vaccine, non-infection cohorts, with each individual having a mean of ~25 days of total monitoring time with an identical patch sensor, to serve as a control group. This cohort was composed of 55 individuals undergoing prospective monitoring for COVID infection and 21 healthcare workers participating in a worker burnout study. Of the 54 individuals with available demographic data, 33% were male with a mean age of 38 (±12.1) and range of 20 to 60 years, making them comparable to the study population. These data were processed in the same way as the current data set to generate MCIRs. From each of these 76 individuals 100 different 72-h periods of MCIR data were randomly selected and tested for the presence of a 6-hour sliding window that met the above definition of "detectable." Each chunk of 72-hour data was labeled as a positive or negative decision depending on whether a 6-h window of MCIR met the definition of detectable. In each of 1000 bootstraps we randomly selected 76 estimated decisions (one per each vaccine dose) and compare them to decisions labeled as negative for the control group and 66 positive from the VIII study cohort to estimate the performance of "detectable response"-identification algorithm and corresponding 95% confidence intervals (CI); True Positive Rate (TPR) = 100%, Specificity (SPC) = 67.7% (67.4%, 68.0%), False Positive Rate (FPR) = 32.3% (32.0%, 32.6%), Positive Predictive Value (PPV) = 73.0% (72.8%, 73.2%), Negative Predictive Value (NPV) = 100%, Acceptance (ACC) = 82.7% (82.5%, 82.9%).

## Statistical analyses

**Population level daily summary changes.** To test for a difference in the pre- and post-vaccination levels for physiologic and behavioral biometrics over the entire study population, medians were calculated using all available pre-vaccine data and 5-day post vaccine data. Statistically significant differences were determined based on Wilcoxon signed rank test.

**Post-vaccine objective reactogenicity by vaccine and individual characteristics.** The probabilities that the two data sets come from different continuous distributions at the 5% significance level are obtained using Kolmogorov–Smirnov (KS) test.

**Relationship of MCIR to subjective reactogenicity.** We compared the AUC MCIR between the two populations—those that reported systemic symptoms and those that reported having no symptoms or local symptoms only—using a two-sample Kolmogorov–Smirnov goodness-of-fit hypothesis test (KS test) to see if there is a significant difference between the two populations in MCIR response. We also carried out similar analyses for each individual parameter (HR, HRV, temperature and RR) to compare their relationship to predicting reactogenicity to that of the multivariate biomarker, MCIR.

**Relationship of MCIR to immunogenicity.** The vaccine-associated change in anti-spike protein IgG and CD4 + /IL-21+ and CD8 + / IFNγ + T-cells was determined by subtracting each participants' levels at 5-days prior to that vaccine dose from their levels at 14-days after vaccination. These vaccine-related changes were compared to AUC MCIR to assess the Spearman correlation between the two. In case of 2nd doses, the baseline values are those obtained prior to vaccine dose two (post 1st

vaccine) while in case of 3rd doses the baseline values were obtained from blood samples drawn prior to third dose (post 2nd vaccine).

One participant was excluded due to insufficient wearable data. The reasoning for removal of another 3 participants' data in evaluating the relationship between AUC MCIR and cellular immunogenicity and 2 participants' data for humoral immunity is explained in Supplementary Fig. 3.

The fitted lines were obtained with a robust fit method which is an alternative regression in the presence of outliers or influential observations making it less sensitive to outliers than standard linear regression. In cases (a) and (b) of Fig. 5 we used Welsch weight function and for (c) we used Andrews weight function, all with the same tune parameter 0.8. The choice of tuning parameter affects the number of data points used in fitting procedure and the model statistics. Each data point is weighted differently based on how it affects the model statistics i.e., based on the magnitude of the residual for that data point.

### Immunogenicity studies
Initial processing of blood samples to isolate peripheral blood mononuclear cells (PBMCs) and blood plasma occurred within 2 h of blood draw. When possible, 1.5 ml of plasma was removed from the top of each spun sample tube prior to buffy coat isolation. Aliquots of plasma were stored at -80℃. PBMCs, with a target concentration of $10.0 \times 10^6$ PBMC/ml/vial, were cryopreserved in liquid nitrogen until batch analysis.

**Flow cytometry-based T-cell assays.** Intracellular cytokine stain assay was performed at CellCarta Bioscience, Inc. (Montreal, QC, Canada) similar to as previously described[24]. For each sample, 4 conditions were used: DMSO, S peptide small pool, non-S peptide pool and staphylococcal enterotoxin B as a positive control. PBMCs were rested and then stimulated for 16–18 h at 37 °C, 5%CO2 in the presence of secretion inhibitors. After the stimulation, cells were stained with fixable Aqua dead cell stain as well as surface antibodies, followed by intracellular staining with cytokine (e.g. IFNγ, IL-2, IL-4, and IL-21) or cytotoxicity marker (perforin) using BD Cytofix/Cytoperm protocol. We selected IL21+ cells as they play a critical role in B-cell activation and correlate with post-vaccine humoral response[25]. "While the circulating level of these cells are low, existing literature supports the ability to accurately measure them and their changes with the methodology used [24,26].

Samples were acquired on a BD Fortessa X20 cytometer and data was analyzed using CellEngine software. The frequency of cytokine-producing antigen-specific T-cells was determined by subtraction of the background cytokine response in unstimulated control samples from the positive response in the samples stimulated with SARS-CoV-2 peptide pools. All negative values after subtraction of background were set to 0. The gating strategy is described in Supplementary Fig. 4 with flow cytometric plots for CD4+/IL21+ cells in Supplementary Fig. 5 and for CD8+/ IFNγ+ cells in Supplementary Fig. 6.

**Anti-spike IgG.** Humoral immunogenicity was determined from plasma samples and defined as SARS-CoV-2 anti-spike IgG titers. Anti-spike IgG concentrations were determined by ELISA (reported as ELISA laboratory units [ELU]/mL) at Nexelis (Laval, QC, Canada)[27]. Final concentrations were determined by calculating the geometric mean of all adjusted concentrations for each samples' dilution by interpolation of the optical density values on the 4-parameter logistic standard curve and adjusted according to their corresponding dilution factor, with maximal dilution of 1-in-5000. For below range samples, a concentration of 0 ELU/mL was assigned when no points fell on the standard curve below the lower limit of quantification of 50.3 ELU/ml.

### Reporting summary
Further information on research design is available in the Nature Portfolio Reporting Summary linked to this article.

## Results
### Participants
A total of 107 individuals consented to participate and included 137 vaccine doses. After excluding individuals with insufficient baseline or post-vaccine data, 88 participants were included in this analysis with a total of 104 vaccine doses. Forty-two (47.7%) were female and the mean age (±SD) for the analyzed population was 37.9 (±13.9) years with a range of 19 to 69 years. Eleven people (12.5%) self-reported prior COVID infection. All participants, but one, received one of the two available mRNA COVID-19 vaccines —Moderna's mRNA-1273 (43 doses), Pfizer-BioNTech's BNT162b2 (48 doses), and for 12 doses participants were not sure which mRNA vaccine they received. One person received the Janssen viral vector vaccine. Response to the first vaccine dose (including the 1 Janssen vaccine recipient) was monitored in 15 participants, the second dose in 44 individuals, and a third dose in 45 people. (See Supplementary Table 1 for participant and vaccine characteristics) Fourteen people provided data around 2 different vaccine doses, and one person for three doses.

### Wearable data
The torso ECG patch was worn for a mean (±SD) of 4.2 (± 2.1) days prior to vaccination and 7.6 (± 3.0) days after. A total of 37,279 h of data were analyzed out of a possible 38,832 h of total patch wear time (96% data availability).

At a population level, small but significant changes relative to individual pre-vaccine baselines were detectable in heart rate (HR), activity, skin temperature, heart rate variability (HRV) and respiratory rate (RR) that deviated from pre-vaccine baseline for ~3 to 4 days after vaccination (Fig. 1).

### Uni-parametric physiologic changes from baseline
Individual differences in changes in single physiologic parameters following vaccination were evaluated by tracking deviations in each z-scored measured parameter relative to each participant's 48-hour pre-vaccine dose baseline period. Among the 85 participants who had data surrounding a second or booster dose, detectable changes greater than one standard deviation relative to an individual's baseline were seen in 28 (33%) of individuals in skin temperature, 24 (28%) in HR, 11 (13%) in RR, 7 (8%) in HRV, and 1 (1%) in activity level. In total, a change in one or more individual parameters of one standard deviation or greater was detected in 46 (54%) of participants following a second or third vaccine dose.

To demonstrate how the relative degree of post-vaccine change in one measure did not necessarily predict the relative degree of change in another parameter in the same individual (e.g. high-temperature change predicting a high HR change), Fig. 2, shows relative changes in HR, HRV, RR and temperature relative to pre-vaccine, with each row an individual's post-vaccine response. Supplementary Fig. 1 further highlights the variability in the onset, duration, and degree of change between individuals in each parameter and how a person's response in one parameter did not necessarily predict their response in another parameter.

### Multivariate Change Index of Reactogenicity
The individual differences in multiple parameters are combined into a Multivariate Change Index of Reactogenicity (MCIR), which is a personalized modeling algorithm that uses coincident heart rate, heart rate variability, respiration rate, activity level and skin temperature derived from the patch device as input and is further detailed in the Methods section. To best quantify an individual's total vaccine-associated reactogenicity, the area under the MCIR curve (AUC MCIR) was developed, which encompasses the duration and degree of all measured physiologic changes in the days following vaccination relative to that person's pre-vaccine baseline (Supplementary Fig. 2).

The first vaccine dose was associated with a less pronounced AUC MCIR response relative to those receiving their 2nd or 3rd dose (Table 1 and Fig. 3a). Sixty-five percent of second or booster doses led to a detectable increase in MCIR after vaccination compared to only 53% of those after a first dose, with 'detectable' as defined in the Table and in the Methods

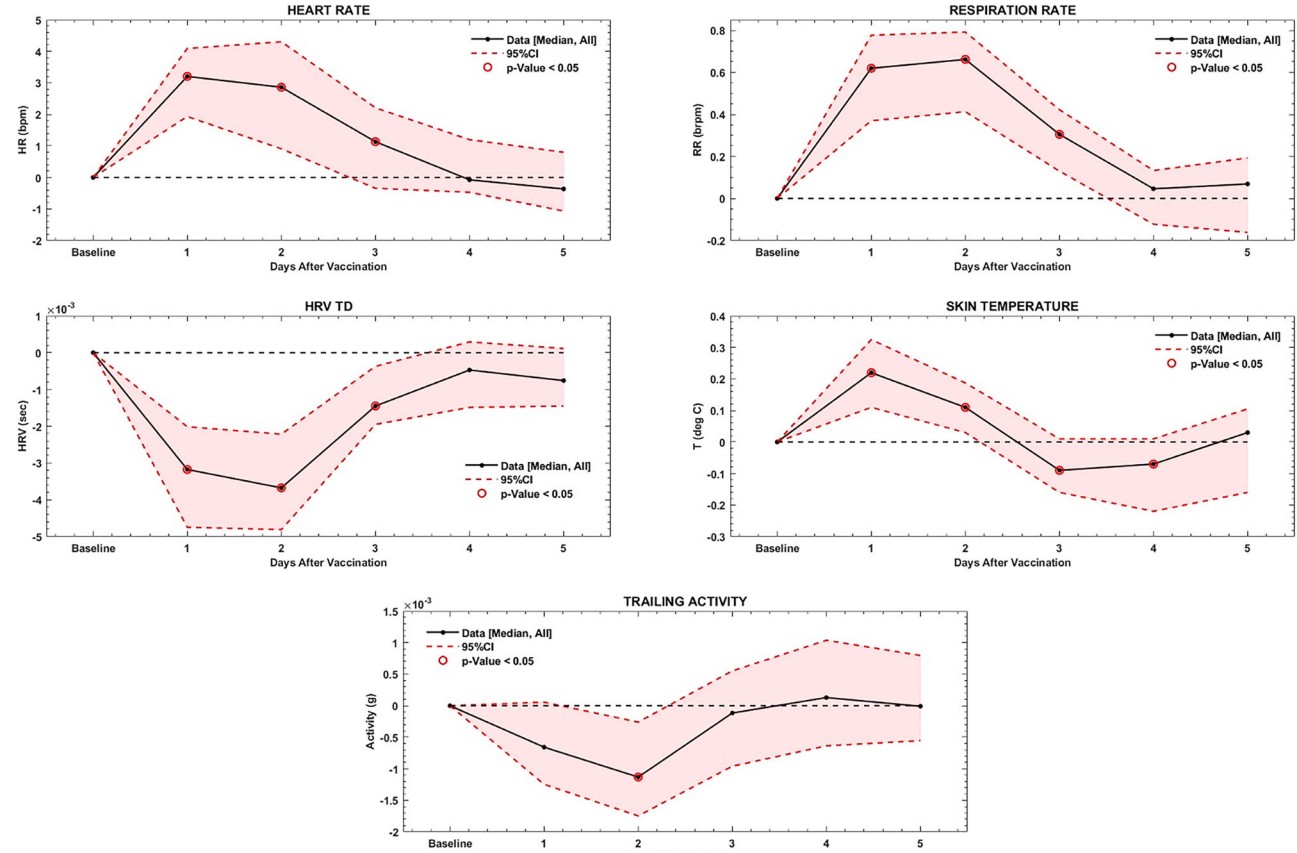

**Fig. 1 | Population level daily summary changes in multiple vital signs and activity following vaccination.** Heart rate (HR), heart rate variability (HRV), respiratory rate (RR), activity level (labeled as "trailing activity" (which indicates that the activity level is filtered using a 20-s moving average at the 1-minute level to remove higher frequency variability over time), and skin temperature trends are shown following any vaccination dose relative to pre-vaccine baseline. Significant changes are circled in red. HR changes are in beats per minute, HRV in seconds, RR in breaths per minute and skin temperature in °C.

section. The total response in participants who received the mRNA-1273 vaccine tended to be greater and appeared to be of longer duration (Fig. 3b) as measured by continuous MCIR, than those treated with BNT162b2, although between group differences were not significant.

### Relationship of MCIR to subjective reactogenicity

Participants were asked to voluntarily self-report any symptoms, or a lack of symptoms, following vaccination via an in-app survey. No data was entered following 15 of the 104 vaccine doses and were excluded from the analysis. Reported symptoms were classified as either systemic or local[4]. Of the 89 doses with post-vaccine symptom data entered, a lack of any symptoms was documented following 10 doses (11.2%), local symptoms only following 9 (10.1%) doses, and systemic symptoms following 70 (78.6%) of doses. Of the 70 doses with systemic symptoms, 50 also reported local symptoms.

Compared to vaccine doses associated with only local or no reported symptoms, those that experienced systemic symptoms had a statistically significantly greater response as measured by AUC MCIR (median [IQR] 0.043 [0.039] vs 0.078 [0.14], $p = 0.008$) (Fig. 4). Similar analyses were carried out for each individual biometric and no significant association was found between post-vaccine HR change ($p = 0.97$), HRV change ($p = 0.33$), temperature change ($p = 0.72$) or RR change ($p = 0.31$) and systemic symptoms.

### Relationship of MCIR to immunogenicity

Twenty-one individuals participated in the immunogenicity sub-study. Their mean age was 37.2 years, 65% were female, and 45% received the mRNA-1273 vaccine, and 55% BNT162b2. Twenty of the 21 sub-study participants had sufficient wearable data to determine their complete

physiologic response following a second or booster vaccine dose. For these sub-study participants, the median AUC MCIR [IQR] was 0.055 [0.15], which was comparable to the overall population. Changes in both T-cell response and SARS-CoV-2 anti-spike protein IgG titers at day 14 post-vaccine relative to their 5-day pre-vaccine levels were compared to the AUC MCIR response following vaccination.

The change in SARS-CoV-2 anti-spike protein IgG titer (ELU/mL) at day 14 following a 2nd or 3rd vaccine dose was significantly correlated with their AUC MCIR after the vaccine (Spearman $\rho = 0.45$, one-sided $p = 0.03$) for the 19 individuals after exclusion of outliers (Fig. 5a). Similarly, the AUC MCIR was directly correlated with the increase in frequency of interleukin-21 expressing CD4+ cells (Spearman $\rho = 0.56$, one-sided $p = 0.009$) at day 14 but was inversely correlated with the change in interferon-gamma expressing CD8+ cells (Spearman $\rho = -0.47$, one-sided $p = 0.029$) for the 17 participants after exclusion of outliers (Fig. 5b, c).

### Discussion

In this work, we characterized the interindividual heterogeneity in physiologic response to vaccination against COVID-19 collected via a medical-grade, wearable, continuous biosensor. Using these data and similarity-based modeling, we developed a digital biomarker, MCIR, that captures the entirety of an individual's unique physiologic response in the days following vaccination relative to their pre-vaccine baseline and demonstrated a significant association between MCIR to the reporting of systemic symptoms, and, in a smaller substudy, humoral and cellular immunity. Reactogenicity, the physical manifestations of the inflammatory response to vaccination, is a significant contributor to vaccine hesitancy, is strongly influenced by the nocebo effect, and may directly correlate with the immune protection

**Fig. 2 | Individual variability in the relationship between the degree of change in specific physiologic parameters following vaccination.** Features averaged over 4 days for each of 70 participants with systemic symptoms and normalized to their pre-vaccine baseline. Relative scaling is [−1 to 0] for negative change and [0 to 1] for positive change for each feature separately.

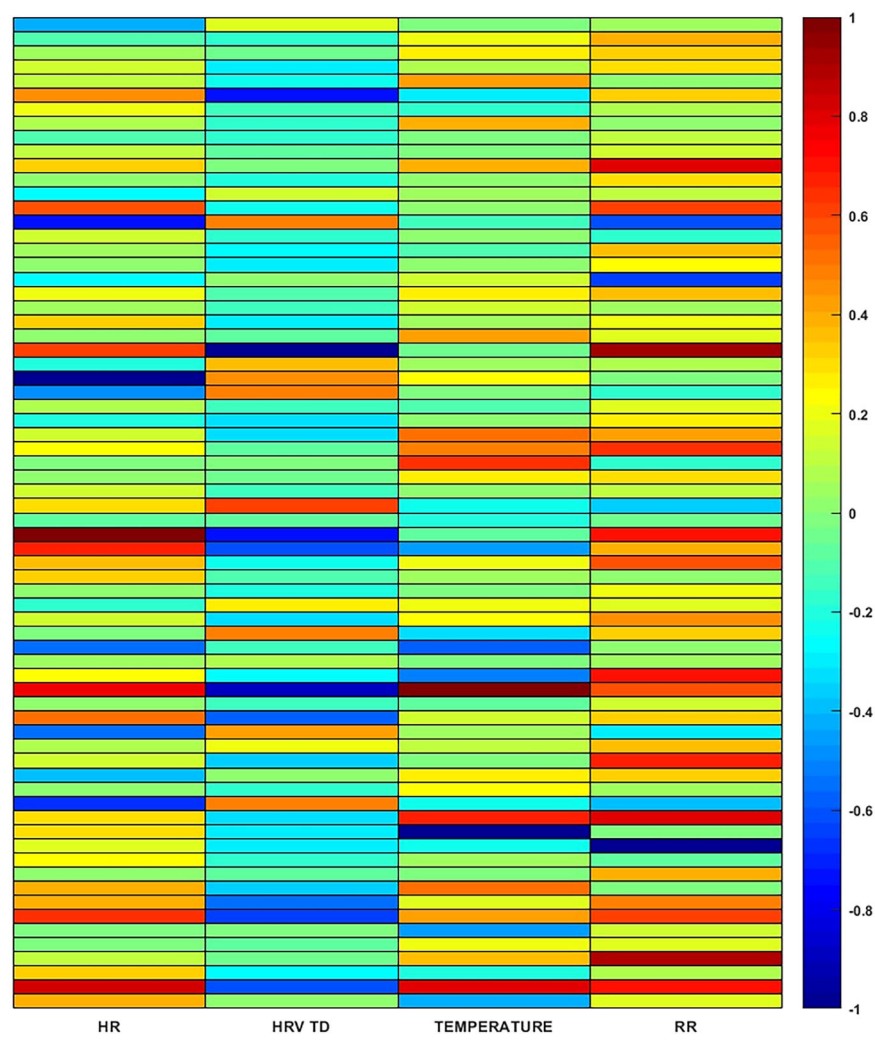

**Table 1 | Total post-vaccine response following 104 vaccines as determined by the Area Under the MCIR curve based on vaccine dose, vaccine type, sex and age**

| | Demographic and vaccine categories | n (%) with a detectable[a] response | Median (IQR) area under the MCIR curve | p-value[b] of comparison of area under the MCIR curve |
|---|---|---|---|---|
| Vaccine Dose | 1st Dose (n = 15) | 8 (53.3) | 0.043 (0.04) | 1st vs 2nd dose: p = 0.01<br>1st vs 3rd dose: p = 0.06<br>2nd vs 3rd dose: p = 0.41 |
| | 2nd Dose (n = 44) | 31 (70.5) | 0.083 (0.18) | |
| | 3rd Dose (n = 45) | 27 (60.0) | 0.063 (0.13) | |
| Vaccine Type[c] | mRNA-1273 (n = 36) | 26 (72.2) | 0.116 (0.19) | p = 0.06 |
| | BNT162b2 (n = 41) | 24 (58.5) | 0.055 (0.096) | |
| Sex[c] | Females (n = 41) | 25 (61.0) | 0.070 (0.15) | p = 0.60 |
| | Males (n = 48) | 33 (68.8) | 0.065 (0.14) | |
| Age by Tertile[c] | Lowest Tertile Age (range 19–28) (n = 31) | 21 (67.7) | 0.070 (0.13) | Lowest vs Middle Tertile: p = 0.99<br>Middle vs Highest Tertile: p = 0.13<br>Lowest vs Highest Tertile: p = 0.08 |
| | Middle Tertile Age (range 28–45.8) (n = 26) | 20 (76.9) | 0.056 (0.12) | |
| | Highest Tertile Age (range 45.8–69) (n = 29) | 16 (55.2) | 0.077 (0.20) | |

[a]Detectable was defined as the presence of at least 50% of 15-min steps within a 6-hour sliding window having a MCIRs > 0.10.
[b]p-values obtained using Kolmogorov–Smirnov (KS) test.
[c]Includes only second doses or higher.

achieved[3,5,28]. These considerations make the ability to objectively measure a person's response to a vaccine important for the enhancement of vaccine development and deployment.

Immune responses are known to vary greatly between people[29]. Consistent with that, in the current study we identified substantial inter-individual variability in the physiologic response to vaccination that was, at least in part, consistent with known differences in vaccination-induced reactogenicity, such as less symptoms following a first dose relative to subsequent doses in people without prior COVID infection, or greater symptoms in those receiving the mRNA-1273 versus BNT162b2 vaccine[4].

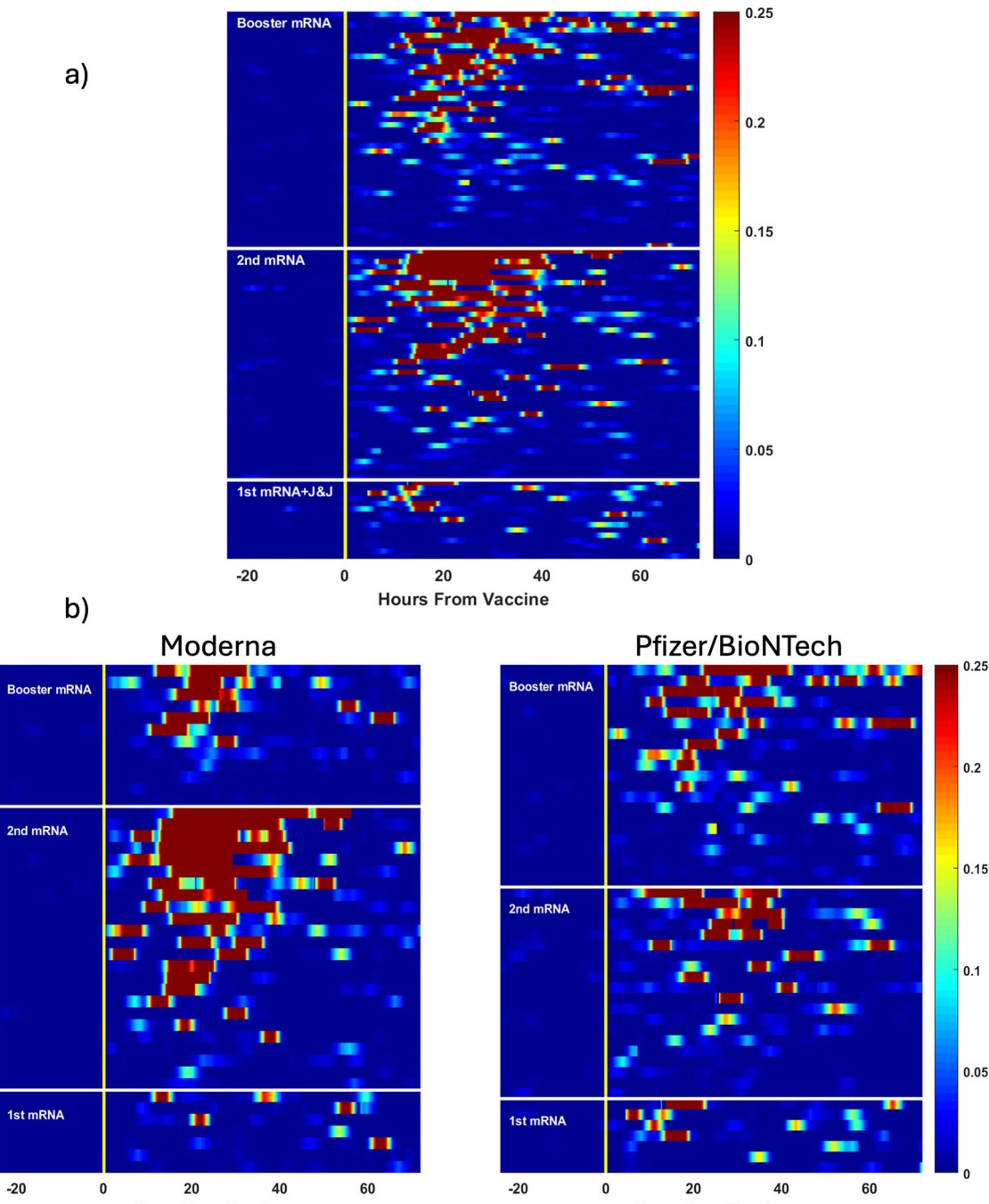

**Fig. 3 | Heatmaps showing inter-individual variation in the onset, degree, and duration of the multivariate change index of reactogenicity (MCIR). a** All 104 vaccine doses, each an individual row, aggregated by first, second or third vaccine dose. **b** Similar heatmaps following 91 vaccine doses, grouped by vaccine type for 43 Moderna and 48 Pfizer/BioNTech vaccine doses.

We too found significantly greater objective reactogenicity, as measured by AUC MCIR, following a second dose relative to a first dose, and, similarly, mRNA-1273 treated individuals had a numerically greater inflammatory response compared with those receiving the BNT162b2 vaccine, although the difference was not statistically significant. While prior studies have found female versus male sex and younger versus older age to be associated with greater reactogenicity[30], we were only able to show similar directionality for a sex difference, but not statistically significant differences in responses for sex or age, likely due to our small sample size.

In our small sub-study, we also found that an individual's AUC MCIR response following vaccination was significantly associated with both humoral and cellular immune response at 14 days after vaccination. These results are consistent with multiple studies that have correlated higher degrees of reactogenicity with a greater immune response[31–34], although other studies have not supported this relationship[35–37]. These inconsistent findings, and the known confounding by the nocebo affect[5], highlight the need for objective measures of reactogenicity. While it is possible to measure serum inflammatory biomarkers to directly determine inflammatory changes after vaccination, due to their invasive nature these studies are limited in size and number. Two such studies in individuals receiving the BNT162b2 vaccine found a correlation between the level of increase of serum inflammatory biomarkers and subsequent Spike antibody levels [7,38].

Multiple prior studies have also shown that subtle, individualized physiologic changes associated with vaccination against COVID-19 are

detectable through wearable sensors. Most of these studies used sensor data from consumer devices, with biometrics typically summarized as a single data point per day. The majority used wrist- or ring-wearable data, evaluating changes post-vaccine in heart rate[12,13,15–17], respiratory rate[13,15], peripheral temperature[13], HRV[13,15–17], sleep[12,15], and activity[12]. One additional study used a multiparametric sensor patch that in addition to the above parameters also included oxygen saturation, blood pressure, cardiac output and systemic vascular resistance[14]. Our study adds to this prior work in several ways. First, we used a medical-grade patch sensor that provided high-fidelity, beat-to-beat data that were processed to produce 17 source signals at a one-minute sampling rate. These continuous data enabled a person's unique physiologic characteristics, and how they vary over time and during different activities, to be precisely defined. Our analytics incorporated all data streams and their simultaneous interactions using a machine learning method of similarity-based modeling to create a 'digital twin' of each participant[39]. This allowed us to continuously compare monitored physiological signals with each participant's baseline model of their unique dynamic physiologic patterns, which effectively removed

expected activity-related, circadian and other personalized variations and left only vaccine-induced differences.

As of August 2024, approximately 5.63 billion people (~71% of the world's population) have received a vaccine against COVID-19[40]. The overwhelming majority of these people received the same dose, or series of doses, depending on the vaccine type, and unrelated to their individual underlying immune state. This is despite the fact that a host of personal characteristics influence the immune response to vaccination including age and sex[41], race[42], genetic and epigenetic factors[43,44], gut microbiome[45], sleep before, and time of day of vaccination[46], previous immune system exposures[47], and much more known and unknown[48]. The inability to objectively measure a person's physical response to vaccination is a major unmet need. Recent multi-omics and high-level transcriptional profiling studies in hundreds of individuals before and after vaccination have confirmed the complexity and broad heterogeneity in immune response to vaccination[44,49]. This complexity in individual response was confirmed in a study involving 820 adults receiving one of 13 different vaccines as nearly two-thirds of the transcriptome variance was unexplained by identifiable clinical or vaccine characteristics [49].

Recognizing that a person didn't experience the expected reponse after a vaccine could potentially influence the timing or frequency of a booster dose. This might be especially important as personalized cancer vaccines continue to be developed[50]. Following further validation, the personalized digital biomarker described here could enable a method for objectively tracking an individual's physical manifestations of their inflammatory response to vaccination, which could help guide vaccine development and eventually, potentially help guide individualized dosing regimens.

As we are detecting physiologic changes in an individual relative to their pre-vaccine normal, it is possible that systemic changes beyond those associated with vaccine-induced inflammation, such as vaccine-induced anxiety, can also contribute to MCIR. Future work that can disambiguate these conflicting contributors will be important. While there are advantages to the real-world study design in terms of eventual implementation, there are also multiple limitations. The largest limitation, especially for the development of a biomarker designed to identify the physical manifestations of vaccine-induced inflammation, is the lack of serum biomarkers to confirm that the physiologic changes detected post-vaccination are due solely to inflammatory changes. Another limitation is that our analysis is restricted to a patch sensor data. While the ECG provides higher quality heart rate data and its derivatives than do wrist- or ring-based photoplethysmography-based sensors, we are unable to determine if the greater data availability improves the clinical value in terms of quantifying reactogenicity or predicting immunogenicity. Although no participant documented that they took an anti-inflammatory agent in post-vaccine period, it is possible that

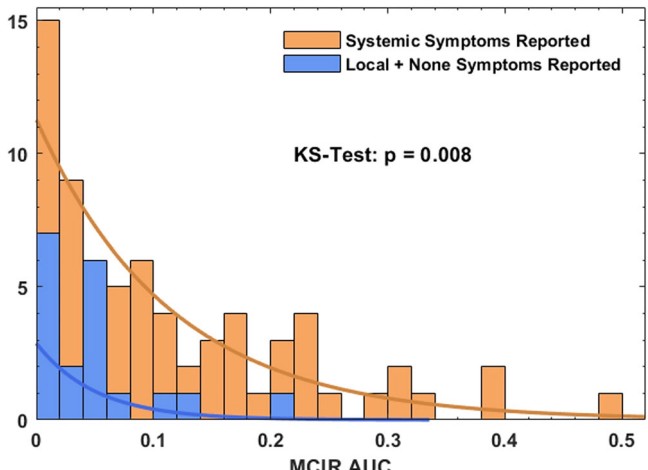

**Fig. 4 | Distribution of individuals' Area Under the Curve (AUC) for the multivariate change index of reactogenicity (MCIR) level and their own self-reported reactogenicity.** Data from 89 doses with reported symptoms classified by whether they were associated with the participant reporting systemic symptoms ($n = 70$) or just local or no symptoms ($n = 19$). KS = two-sample Kolmogorov–Smirnov goodness-of-fit hypothesis test.

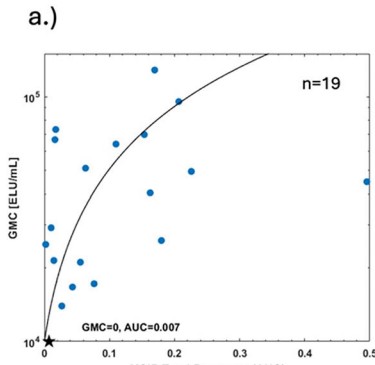
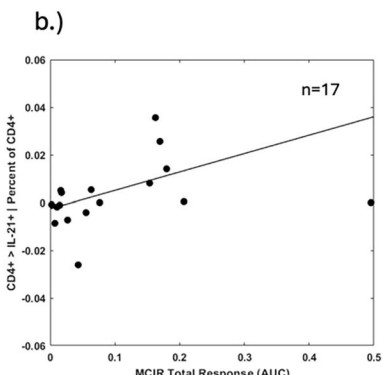
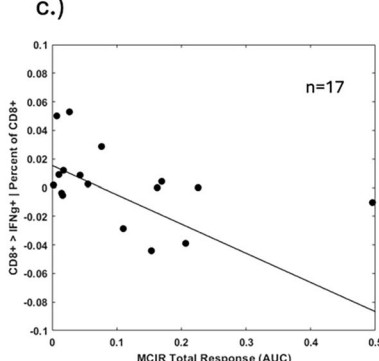

**Fig. 5 | Relationship between an individual's area under the curve (AUC) for the multivariate change index of reactogenicity (MCIR) and their immunogenicity. a** AUC MCIR total response versus change from baseline of Anti-SARS-CoV-2 anti-spike protein IgG titer in 19 participants following a 2nd or 3rd mRNA vaccine dose. The individual designated by the star had a lower titer at day 14 than at baseline.

**b** AUC MCIR total response versus change from baseline of frequency of interleukin-21 (IL-21+) expressing CD4+ cells and (**c**) MCIR AUC total response versus change from baseline of frequency of interferon-gamma (IFN-γ+) expressing CD8+ cells in 17 participants 14 days following 2nd or 3rd mRNA vaccine dose.

some did, and we were unable to account for their effects. Another potential limitation is that we were limited to analyzing vaccine-induced cellular immunity changes in only 2 T-cell subpopulations—CD4 + /IL-21+ and CD8 + /IFNγ. Finally, the inverse relationship between MCIR-detected reaction and interferon-gamma expressing CD8+ cells is difficult to explain physiologically, which might suggest it is a chance finding.

There is an unmet need for a noninvasive method to objectively measure the totality of reactogenicity. In this work, we describe the development of a personalized digital biomarker for vaccine-induced reactogenicity by combining medical-grade wearable sensor data and machine learning-enabled digital twin technology in the setting of real-world vaccination against COVID-19. We found that our digital biomarker, MCIR, correlated with subjective reactogenicity better than any single physiologic parameter and, in a small substudy, with immunogenicity. If confirmed in further studies, a personalized digital biomarker for reactogenicity could play an important role in improving vaccine safety and efficacy.

## Data availability
Data used to develop the Figures is available in the provided Supplementary Data files 1 through 5. Requests for de-identified summary data from this study can be made by submitting a written request with an analysis plan to the corresponding author for review. Only de-identified individual-level data may be shared to protect participant privacy, and contingent on IRB approval.

## Code availability
Machine learning algorithms and the Prolaio platform are proprietary, and FDA cleared. The Prolaio platform, including the MCIR and other biomarker algorithms, can be accessed through partnership with Prolaio, Inc. Researchers interested in collaboration should contact the corresponding author for further information.

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

## Acknowledgements

This study was sponsored by physIQ, Inc. (Chicago, IL, USA) a company that no longer exists and whose assets were purchased by Prolaio, Inc. (Scottsdale, AZ, USA).

## Author contributions

Conception or design of the work: S.R.S., J.S., S.W., C.J.G. Acquisition of data: M.G., H.G., J.L.A., E.P. Analysis and/or interpretation of data: M.P.W., S.A., D.W., J.S., S.W., H.G., E.P., S.R.S. Drafting, review, and critical revision of the manuscript: S.R.S., J.S., M.G., H.G., M.P.W., C.J.G., J.L.A., S.A., D.W., E.P., S.W.

## Competing interests

The current study was sponsored by physIQ, Inc. (Chicago, IL, USA) a company that has since been purchased by Prolaio, Inc. (Scottsdale, AZ, USA). A patent was filed by physIQ for the digital biomarker described (US Patent Application No. 18/226,408). Some co-authors are employees of Prolaio, Inc. (J.S., M.G., S.W.). H.G. and E.P. are employees of CellCarta Bioscience, Inc. (Montreal, QC, Canada). Other authors (S.R.S., C.J.W., J.L.A., M.P.W., S.A., D.W.) report no competing interests.
