## [Transparent Peer Review file · Communications Medicine]

Wearable Sensor and Digital Twin Technology for the Development of a Personalized Digital Biomarker of Vaccine-Associated Reactogenicity

Corresponding Author: Dr Steven Steinhubl

Version 0:

Reviewer comments:

Reviewer #1

(Remarks to the Author)

Currently, measurement of post-vaccination inflammation or reactogenicity either relies on self-reporting of symptoms or by analysing the level of inflammatory mediators in a blood draw. If such reaction could be measured in an unbiased and non-invasive manner, it would have significant impact on vaccine research and the management of communicable diseases in general.

In this manuscript, Steinhubl et al. reported a torso ECG patch that detected consistent trend of transient physiological changes among participants over the 3-day period after receiving COVID-19 vaccines. Of note, the authors have developed a machine learning method that normalised post-vaccine responses that were collected over several days, against individualised baseline collected over several days prior to vaccination. Such normalisation reduces background noise and unmask otherwise subtle vaccine-induced changes. Another strength of the study is the use of an inflammation Multivariate Change index (iMCL) that summarises physiological changes measured over five parameters into a single score that can be compared longitudinally as well as between participants.

One major concern that significantly undermines the meaningfulness of this study is, while the study premises is very interesting and could be highly impactful, there is little evidence that the biosensor were actually measuring inflammatory responses. Figure 1 and 2 showed that the biosensor measured heart rate, heart rate variability, respiration rate, skin temperature and activity level. While these parameters can indeed change in response to inflammation, they are, on their own, not direct indicators of inflammation. A strong correlation between what the biosensor measures and the level of inflammatory mediators (e.g. cytokines/chemokines or prostaglandins) would be essential for substantiating the central claim that the biosensors measured inflammatory responses.

Another major concern is with the immunogenicity sub-study show in Figure 5. Since participants were analysed receiving their second or third vaccine dose, one would expect some level of pre-existing antibody and T cells that can variable markedly between participants. Without normalising against each participant's own pre-vaccine antibody or T cell response, it'd be impossible to conclude on immunogenicity associated with the most recent vaccine dose, or calculate how it correlates with changes measured by the biosensor, which relates to the most recent vaccine dose only. Using IL-21 as the sole indicator of antigen-specific CD4+ T cell responses following re-stimulation is a somewhat unusual choice and should be appropriately justified. Within either CD4+ or CD8+ T cell compartment, the recall response appears to be extremely low (<0.05% of total CD4+ or CD8+ T cells), especially at day 14 after the second or third vaccine dose when the T cell responses would be expected to be at the peak. The reported T cell responses is at least 100-1000 folder lower than other studies would suggest.

Reviewer #2

(Remarks to the Author)

Wearable Sensor and Digital Twin Technology for the Development of a Personalized Digital Biomarker of Vaccine-Induced Inflammation

In their manuscript, the authors describe the development of personalized digital biomarkers by integrating data from VitalPatch™, a wearable sensor, with machine learning methods. The device continuously collected data on heart rate, heart rate variability, respiration rate, activity level, and skin temperature before and after COVID-19 vaccination. They then applied machine learning algorithms to detect changes in inflammatory responses following vaccination.

Vaccination stands as one of the greatest achievements of medical science which annually saving millions of lives

worldwide. However, vaccine efficacy varies significantly at the individual level and thus not everyone is equally protected when vaccinated by a particular vaccine. Assessing vaccine response at the personal level can help to determine individual risk of having a particular disease following vaccination. Yet, monitoring vaccine responses at personal level is challenging and thus rarely practiced.

Advances in AI technology and wearable devices for continuous monitoring of physical and biological parameters provides the hope to monitoring of immune responses of each vaccinated person in near future. Steinhubl et al.'s work on wearable sensors and digital twin technology to monitor vaccine-induced inflammation at individual level thus timely and valuable. While this work represents a significant step towards to monitor vaccine immune responses at a personal level, the authors consider the following issues:

1. Study Design

a. The authors mentioned that they used data from controls to estimate the false positive rate of the algorithm. However, they have not provided any further details about the control selection process, data collection process, and whether those controls were comparable to the vaccinated people included in this study. A comparative table about the control and exposed groups would ensure data clarity.

2. Analysis

a. The authors have not demonstrated whether they included any metadata, such as age, sex, occupation, or usage of analgesic drugs, in the iMCI model, which may influence the outcome.

b. In the current study, most of the data were collected around the second and subsequent doses of the vaccine. Therefore, a sensitivity analysis should be done to ensure that the results would be similar if information related to the first dose of the vaccine were collected from the majority of the participants.

3. Results

a. While vaccine reactogenicity correlates with host factors such as age, sex, vaccine doses, and usage of analgesic drugs, the basic characteristics of the study participants are missing, making it difficult to interpret the study outcomes. Sharing the basic characteristics of the participants in a tabular format might help readers interpret the study findings.

4. Discussion

a. The authors reported a small difference in the physiological changes related to vaccine reactogenicity. They should discuss whether this is not due to artifacts such as vaccinated persons experiencing more anxiety, remaining in a resting position after vaccination due to fear of illness, or using analgesic drugs.

b. The authors should elaborate on why they found a less detectable response to the first dose of the vaccine, as previous studies have demonstrated a stronger reactogenicity related to the first dose of the vaccine.

Version 1:

Reviewer comments:

Reviewer #1

(Remarks to the Author)

Thanks to the authors for their clear and effective rebuttal responses, which makes the assessment task easier.

Understandably there are practical constraints with human-based studies that going back to perform more analyses is not always possible. In the first round of review process I have raised two major concerns. My review of the authors' response to these two concerns are below.

1. that the biosensor did not measure inflammatory responses to the vaccine directly. The authors have responded very well and appropriately to the comment. The changes were extensive, but I now find the conclusion can be upheld with more confidence. There are, however, a number of mentions of inflammation/inflammatory responses through the manuscript after the revision; the vast majority are appropriate, but the authors may wish to check the mentions in line 176, 260 and 282 to see if inflammatory or reactogenic would be the more appropriate call.

2. whether the antibody and T cell analyses performed were sufficient to address immunogenicity. I accept the authors' explanation and no further analysis is feasible. In order to assess the validity of data around antibody and T cell responses to the spike proteins, by myself and by the readers, I would argue strongly the original analyses should be included as supplementary figure(s). This is because the data presented in Figure 5 had undergone several rounds of calculations already. As a minimum, the ELISA data for antibody responses (e.g. dilution factor used for calculation), representative FACS plots for T cell response including controls, general gating strategy, and T cell data should be shown, and where possible, for each participant.

Reviewer #2

(Remarks to the Author)

I would like to express my gratitude to the authors for sharing the revised manuscript on measuring vaccine-associated reactogenicity using wearable sensors and digital tools. I have read the revised manuscript with great interest, as well as the authors' responses to the previous queries. I feel that the authors have made significant changes to the manuscript that adequately address my concerns. Additionally, I believe this manuscript holds substantial value in guiding future research on innovative technologies to monitor vaccine-related responses, which is especially critical for preterm infants and immunocompromised individuals.

Response to Reviewers

Reviewer 1:

One major concern that significantly undermines the meaningfulness of this study is, while the study premises is very interesting and could be highly impactful, there is little evidence that the biosensor were actually measuring inflammatory responses. Figure 1 and 2 showed that the biosensor measured heart rate, heart rate variability, respiration rate, skin temperature and activity level. While these parameters can indeed change in response to inflammation, they are, on their own, not direct indicators of inflammation. A strong correlation between what the biosensor measures and the level of inflammatory mediators (e.g. cytokines/chemokines or prostaglandins) would be essential for substantiating the central claim that the biosensors measured inflammatory responses.

We completely agree with the reviewer and realize, in retrospect, that we were overly optimistic in claiming a physiologic biomarker for inflammation. While our ultimate desire is to develop and refine a digital biomarker for inflammation, since the results described are from a real-world study we don't have the ability to go back and measure serum inflammatory biomarkers. In addition, the study's funder no longer exists, so continuing the study to gather post-vaccination inflammatory biomarkers is also not possible.

Therefore, to address this major shortcoming we have substantially modified all aspects of the manuscript, from the title to the conclusions, to de-emphasize inflammation and instead highlight the relationship between the developed objective biomarker and subjective reactogenicity.

It is not feasible to point out all of the changes throughout the manuscript in this response, but we will highlight several key changes:

- The title is now more accurate - *Wearable Sensor and Digital Twin Technology for the Development of a Personalized Digital Biomarker of Vaccine-Associated Reactogenicity*
- In the conclusion of the Abstract we now state *"...to measure the degree and duration of physical changes an individual experiences following vaccination. We found that the multivariate digital biomarker better predicted systemic reactogenicity than any one physiological data type and correlated with vaccine-induced changes in humoral and cellular immunity in a 20-person subset."*
- At the conclusion of the Introduction: *"These results provide preliminary support for the potential value of a multivariate digital biomarker to objectively quantify the totality of individual reactogenicity following vaccination."*
- In the Results section, we carried out additional analysis to better demonstrate the performance of a digital multivariate biomarker, now renamed the multivariate change index of reactogenicity (MCIR), relative to individual physiologic parameters, and participant-reported reactogenicity. These analyses include:
 - To better show that the degree of individual change in one biometric (e.g. HR) doesn't necessarily correlate with another we developed a new Figure 2 and added the following description in the Results section.

To demonstrate how the relative degree of post-vaccine change in one measure does not necessarily predict the relative degree of change in another for an individual (e.g. high temperature changes leads to a high HR change), **Figure 2**, shows relative changes in HR, HRV, RR and temperature, with each row an individual's post-vaccine response.

- To demonstrate the performance of a multivariable digital marker relative to any one single physiologic measure we duplicated the MCIR analysis for each measure and, unlike with MCIR, found no significant association with systemic symptoms versus no systemic symptoms. The following sentence was added to the Results section, in the subsection, Relationship of MCIR to Subjective Reactogenicity:

Similar analyses were carried out for each individual biometric and no significant association was found between post-vaccine HR change ($p=0.97$), HRV change ($p=0.33$), temperature change ($p=0.72$), or RR change ($p=0.31$) and systemic symptoms.

Another major concern is with the immunogenicity sub-study show in Figure 5. Since participants were analysed receiving their second or third vaccine dose, one would expect some level of pre-existing antibody and T cells that can variable markedly between participants. Without normalising against each participant's own pre-vaccine antibody or T cell response, it'd be impossible to conclude on immunogenicity associated with the most recent vaccine dose, or calculate how it correlates with changes measured by the biosensor, which relates to the most recent vaccine dose only. Using IL-21 as the sole indicator of antigen-specific CD4+ T cell responses following re-stimulation is a somewhat unusual choice and should be appropriately justified. Within either CD4+ or CD8+ T cell compartment, the recall response appears to be extremely low (<0.05% of total CD4+ or CD8+ T cells), especially at day 14 after the second or third vaccine dose when the T cell responses would be expected to be at the peak. The reported T cell responses is at least 100-1000 folder lower than other studies would suggest.

We appreciate the reviewer pointing out several areas that needed improvement and we could have communicated better.

Regarding baseline normalization, we completely agree. The original analyses and results reported and demonstrated in Figure 5 are normalized to their levels determined ~5 days prior to that specific vaccine dose. In order to make this clearer to readers we have made edits in several locations:

- Results section: Sentence has been modified to state: "*Changes in both T-cell response and SARS-CoV-2 anti-spike protein IgG titers at day 14 post-vaccine relative to their 5-day pre-vaccine levels were compared to the AUC MCIR response following vaccination.*"
- Methods:

- Immunogenicity Sub-study – A sentence was added at the end of the section stating, *"Immune response was determined based on the change from the 5-day pre-vaccine level to the 14-day post-vaccine level."*
- Statistical Analysis/Relationship of MCIR to Immunogenicity – The first sentence has been rewritten to make it clear that 14-day post-vaccine levels of anti-spike protein IgG and T-cell populations of interest were determined relative to their 5-day pre-vaccine levels. *"The vaccine-associated change in anti-spike protein IgG and CD4+/IL-21+ and CD8+/IFN γ + T-cells was determined by subtracting each participants' levels at 5-days prior to that vaccine dose from their levels at 14-days after vaccination."*

Regarding our selection of measuring CD4+/IL-21+ as one of our two measures of change in vaccine-induced cellular immunity, we agree there could be an advantage to measuring multiple markers, but this wasn't possible due to limited resources. We explained this choice selected IL-21+ as our measure of CD4+T-cell response by adding this sentence to the Methods section under Flow cytometry-based T-cell assays: *"We selected IL21+ cells as they play a critical role in B-cell activation and correlate with post-vaccine humoral response."⁴⁹*

We also added a sentence to the Limitations section stating that *"Another potential limitation is that we were limited to analyzing vaccine-induced cellular immunity changes in only 2 T-cell subpopulations - CD4+/IL-21+ and CD8+/IFN γ ."*

Finally, in relation to the low response rate of, this was anticipated as IL21-producing cells tend to leave the circulation and enter lymph nodes when they become activated. However, the literature supports that there are adequate numbers circulating to accurately measure them and changes associated with vaccination. We added this sentence to Flow cytometry-based T-cell assays in the Methods section to explain this: *"While the circulating level of these cells are low, existing literature supports the ability to accurately measure them and their changes with the methodology used."^{48,50}*

Reviewer 2:

1. Study Design

- a. The authors mentioned that they used data from controls to estimate the false positive rate of the algorithm. However, they have not provided any further details about the control selection process, data collection process, and whether those controls were comparable to the vaccinated people included in this study. A comparative table about the control and exposed groups would ensure data clarity.

We agree with the reviewer and should have done a better job describing the control population. As described below in more detail, we have limited demographic data on the study population which limits our ability to carry out detailed comparison to the control population. We have added the following sentences describing the control population in the Methods section under MCIR Detectable Response: *"This cohort was composed of 55 individuals undergoing prospective monitoring for COVID infection and 21 healthcare workers participating in a worker burnout study. For the 54 individuals with available demographic data, 33% were male with a mean age of 38 (+12.1) and range of 20 to 60 years, making them comparable to the study population."*

2. Analysis

a. The authors have not demonstrated whether they included any metadata, such as age, sex, occupation, or usage of analgesic drugs, in the iMCI model, which may influence the outcome.

The model did not include any data beyond the sensor based physiologic data. We agree with the reviewer that age and sex, on a population level, are shown to associated with vaccine response, both reactogenicity and immunogenicity. However, our model is built solely on the specific individual's physiologic data pre-vaccine and their post-vaccine changes relative to their unique baseline. The goal of our model is to move beyond the historic population-based modeling and focus solely on each individual's unique physiology. We have attempted to explain this more clearly in multiple locations in the manuscript, as in this example from the abstract: *"By using each individual's pre-vaccine digital twin, we are able to effectively control for expected physiologic variations unique to that individual in the post-vaccine period, leaving only vaccine-induced differences. We used these individualized differences between the pre- and post-vaccine period to develop a multivariate digital biomarker for objectively measuring the degree and duration of physical changes each individual experiences following vaccination."*

On the other hand, the use of anti-inflammatory analgesics post vaccine would be expected to influence both the symptoms and inflammatory response a person experiences following a vaccine. and we did not adequately document this in the manuscript but have corrected that. We did ask participants to document if they used any anti-inflammatory analgesic medications in the post-vaccine period, but no

participant entered that information. We feel it is unlikely that no participant took an anti-inflammatory agent, so we have added the following sentence to the Limitations section: *"Although no participant documented that they took an anti-inflammatory agent in post-vaccine period, it is possible that some did, and we were unable to account for their effects."*

b. In the current study, most of the data were collected around the second and subsequent doses of the vaccine.

Therefore, a sensitivity analysis should be done to ensure that the results would be similar if information related to the first dose of the vaccine were collected from the majority of the participants.

We agree with the reviewer that while we have stated that the developed biomarker correlates with post-vaccine reactogenicity, the majority of the data supporting that is based on the typically more robust response to a second or third COVID-19 mRNA vaccine dose.

Of the 15 people who received a first vaccine dose during the study, 12 provided data to allow for correlation with reactogenicity, and 10 reported symptoms – 9 systemic and 1 local only. These individuals were included in the analysis of the relationship between AUC iMCI (now MCIR) and reactogenicity symptoms. In this small subset the relationship between MCIR and the presence of systemic symptoms was similar to the overall study findings but not statistically significant (KS $p=0.083$) The relationship is shown in the figure, but we have not added it to the revised manuscript.

3. Results

a. While vaccine reactogenicity correlates with host factors such as age, sex, vaccine doses, and usage of analgesic drugs, the basic characteristics of the study participants are missing, making it difficult to interpret the study outcomes. Sharing the basic characteristics of the participants in a tabular format might help readers interpret the study findings.

As noted above, we collected a limited amount of demographic data as our goal was not to explore population based differences but rather identify individual changes. We have tried to include all the pertinent information we collected in narrative form, but to make it clearer to readers we have generated the Table below and have added it to the Supplementary Information:

Participant Characteristics (n=88 participants, 104 vaccine doses)	
Mean Age (+SD)	37.9 (+13.9)
Female n (%)	42 (47.7)
Self-Reported Prior COVID Infection n (%)	11 (10.6)
Received an initial vaccine dose n (%)*	15 (14.4%)
Received a second vaccine dose n (%)*	44 (42.3)
Received a third vaccine dose n (%)*	45 (43.3)
Received a dose of Moderna's mRNA-1273 n (%) of 92 who knew type	43 (46.7)
Received a dose of Pfizer-BioNTech's BNT162b2 n (%) of 92 who knew type	48 (52.2)

*Total percentages add up to >100% as 14 participants provided data for 2 doses and 1 person for 3 doses

4. Discussion

a. The authors reported a small difference in the physiological changes related to vaccine reactogenicity. They should discuss whether this is not due to artifacts such as vaccinated persons experiencing more anxiety, remaining in a resting position after vaccination due to fear of illness, or using analgesic drugs.

We agree with the reviewer that while we are detecting individualized changes post-vaccine relative to their pre-vaccine normal, it is certainly possible that reactions beyond those directly associated with vaccine-induced inflammation, like anxiety, can potentially contribute to those changes. We have added the following sentence to the beginning of the Limitations section of the Discussion: *"As we are detecting physiologic changes in an individual relative to their pre-vaccine normals, it is possible that systemic changes beyond those associated with vaccine-induced inflammation, such as vaccine-induced anxiety, can also contribute to MCIR. Future work that can disambiguate these conflicting contributors will be important."*

b. The authors should elaborate on why they found a less detectable response to the first dose of the vaccine, as previous studies have demonstrated a stronger reactogenicity related to the first dose of the vaccine.

Other than for individuals with prior COVID-19 infection (which was only one of our 15 participants who were monitored at the time of their first vaccine dose), we believe that the data supports that reactogenicity was significantly less in individuals following a first dose relative to subsequent doses. The fact that our findings regarding MCIR mirroring that result is encouraging. For example, from V-Safe data of >3.6 million individuals receiving an mRNA vaccine against COVID-19, systemic symptoms occurred in 51.7% of individuals after a first dose

compared with 69.4% after the second dose. (Reference #4 from the manuscript, JAMA. 2021;325(21):2201-2202. doi:10.1001/jama.2021.5374)

COMMSMED-24-0086A

Response to Reviewers

Reviewer 1:

1. That the biosensor did not measure inflammatory responses to the vaccine directly. The authors have responded very well and appropriately to the comment. The changes were extensive, but I now find the conclusion can be upheld with more confidence. There are, however, a number of mentions of inflammation/inflammatory responses through the manuscript after the revision; the vast majority are appropriate, but the authors may wish to check the mentions in line 176, 260 and 282 to see if inflammatory or reactogenic would be the more appropriate call.

Thank you for pointing out these 3 instances. We made changes in each of these specific instances in lines 176, 260, and 282 of the previous revised manuscript. We either just deleting the word "inflammation" or replaced it with a more appropriate substitute, such as reactogenicity or reaction.

In addition, we performed a word search for inflammation and inflammatory throughout the manuscript and found 4 additional locations, 3 in the Methods section and one in the legend for Table 1, where we felt a different term would be more appropriate and made changes here too.

2. Since whether the antibody and T cell analyses performed were sufficient to address immunogenicity. I accept the authors' explanation and no further analysis is feasible. In order to assess the validity of data around antibody and T cell responses to the spike proteins, by myself and by the readers, I would argue strongly the original analyses should be included as supplementary figure(s). This is because the data presented in Figure 5 had undergone several rounds of calculations already. As a minimum, the ELISA data for antibody responses (e.g. dilution factor used for calculation), representative FACS plots for T cell response including controls, general gating strategy, and T cell data should be shown, and where possible, for each participant.

We appreciate the reviewer pointing out the value of this additional information for readers.

To better explain the antibody response, we have added more information in the Methods section in the "Anti-spike IgG" paragraph. Rather than just deferring to prior similar work by the same lab using the same ELISA technique described in Reference #51, we added the following 2 sentences:

Final concentrations were determined by calculating the geometric mean of all adjusted concentrations for each samples' dilution by interpolation of the optical density values on the 4-parameter logistic standard curve and adjusted according to their corresponding dilution factor, with maximal dilution of 1-in-5000. For below

range samples, a concentration of 0 ELU/mL was assigned when no points fell on the standard curve below the lower limit of quantification of 50.3 ELU/ml.

In addition, to better clarify the cellular immunity response work, three figures with corresponding explanatory legends have been added to the supplementary material:

- Supplemental Figure D explains the gating strategy.
- Supplemental Figures E and F are the FACS plots for all participants for CD4+ > IL21+ as percent of CD4+, and for CD8+ > IFN γ + as percent of CD8+, respectively.

These Supplemental Figures are called out in the Methods section in the section "Flow cytometry-based T-cell assays."

Reviewer 2:

I would like to express my gratitude to the authors for sharing the revised manuscript on measuring vaccine-associated reactogenicity using wearable sensors and digital tools. I have read the revised manuscript with great interest, as well as the authors' responses to the previous queries. I feel that the authors have made significant changes to the manuscript that adequately address my concerns. Additionally, I believe this manuscript holds substantial value in guiding future research on innovative technologies to monitor vaccine-related responses, which is especially critical for preterm infants and immunocompromised individuals.

We very much appreciate the Reviewer's kind comments, regarding the revisions and its potential value.